# Microstructure, Mechanical Properties and Wear Behavior of High-Velocity Oxygen-Fuel (HVOF) Sprayed (Cr$_3$C$_2$-NiCr+Al) Composite Coating on Ductile Cast Iron

**Marzanna Ksiazek** [1,*]**, Lukasz Boron** [2] **and Adam Tchorz** [2]

[1] Department of Non-Ferrous Metals, AGH University of Science and Technology, 30 Mickiewicza Ave., 30-059 Cracow, Poland

[2] Łukasiewicz Research Network- Foundry Research Institute, 73 Zakopianska St., 30-418 Cracow, Poland; lukasz.boron@iod.krakow.pl (L.B.); adam.tchorz@iod.krakow.pl (A.T.)

[*] Correspondence: mksiazek@agh.edu.pl

**Abstract:** In the present work Cr$_3$C$_2$-NiCr powder containing Al particles was deposited on ductile cast iron with high-velocity oxy-fuel (HVOF) thermal spray coating technique. An investigation was conducted to determine the role of Al particles in the Cr$_3$C$_2$-NiCr coating produced with HVOF technique on microstructure, mechanical and wear properties in a system Cr$_2$C$_3$-NiCr coating/ductile cast iron. The microstructure of the HVOF-sprayed Cr$_3$C$_2$-NiCr+Al coating was characterized by light microscopy, X-ray diffraction (XRD), scanning electron microscope (SEM), transmission electron microscope (TEM) and energy dispersive X-ray spectroscopy (EDS). Microstructure analysis reveals the formation of coating with low porosity, good adhesion to the substrate and dense structure with irregularly shaped particles of Al arranged in strips and finely fragmented Cr$_3$C$_2$ particles embedded in a nanocrystalline Ni-Cr alloy matrix. In addition, the results were discussed in reference to examination of bending strength considering cracking and delamination in the system of (Cr$_3$C$_2$-NiCr+Al)/ductile cast iron as well as microhardness and wear resistance of the coating. It was found that the addition of Al particles significantly increased resistance to cracking and wear behaviour in the studied system.

**Keywords:** Cr$_3$C$_2$-NiCr coating; thermal spraying; HVOF; wear resistance

---

## 1. Introduction

Currently, supersonic coatings are widely used in various industries to increase the resistance of machine parts to abrasive wear, erosion, corrosion, etc., which consequently results in a significant increase in durability and reliability of machine parts, reduction of production costs and saving energy and materials [1,2]. An important application of coatings sprayed with the method of HVOF (High-Velocity Oxy-Fuel) is as working surfaces of turbine blades in power installations; shaft necks; pump impellers; bodies; slide bearings; sleeves; transport rollers; heat exchangers; guides and power hydraulic elements, e.g., plungers, actuators, piston rods, valves. Of the various carbides, the most extensive use has been found in chromium and tungsten carbide [3–6]. The HVOF spray method allows the execution of coatings with special properties such as very low porosity, high adhesion, low oxygen content and with a limitation of the carbide decay process, decarburization and carbide oxidation. The coating in this process is formed primarily of particles in high-plastic state (T in the range of 1600–2000 °C), because their residence time in the stream of gases is very short. The high kinetic energy of the particles, even though they are not in a liquid state, allows them to be deformed accordingly,

resulting in a coating that is very well bound to the substrate with a minimum porosity. The HVOF process uses high kinetic energy of the particles having relatively low temperature; therefore, the grain growth is restricted, and it is possible to obtain coatings with highly fragmented grain. At the same time, high velocity (400–800 m/s) limits the harmful effect of oxygen on the microstructure of powder grain during the flight, and its impact on the coating being formed [7]. In particular, the coating produced in the HVOF process, consisting of hard $Cr_3C_2$ grains, which are embedded in the nickel-chromium alloy matrix, is characterized by very good resistance to wear at high temperature, even up to 800 °C [8]. This coating microstructure is advantageous because fine and hard $Cr_3C_2$ particles provide high hardness and wear resistance of coatings, and the relatively soft material of the NiCr matrix provides high impact strength. In addition, the extensive fragmentation and even distribution of carbide particles in the matrix favourably affect the physical and mechanical properties. The perspective direction in a surface engineering is also the development of methods for the production of high-quality composite coatings based on chromium and tungsten carbides, providing high resistance to the abrasion and to the chemical impact of an environment [9].

The aim of the study was to evaluate the effect of modifying chemical composition, by doping the standard powders $Cr_3C_2$-25NiCr with Al metal particles during consolidation of coatings on ductile cast iron, in the process of high velocity supersonic spraying of powder (HVOF), on microstructure, physic-mechanical and wear properties of the system of a composite coating *($Cr_3C_2$-NiCr+Al)*/ductile cast iron, increasing the mechanical durability of the *coating-substrate* system.

## 2. Experimental Procedure

The composite coating was produced by supersonic powder spraying of carbide powder with $Cr_3C_2$-25%NiCr composition with a nominal grain size distribution of –45 + 15 μm and a carbide size of 5–20 μm (Diamalloy 3004 Salzer-Metco; Pfaffikon, Switzerland), into which 10 wt % of 20 μm Al particles were introduced. In order to spray the coatings, a HV50 HVOF system in the company Plasma System S.A. (Siemianowice, Poland) was used, where a mixture of kerosene and oxygen was used as the fuel for the spraying process. The substrate made of ductile cast iron EN-GJS-500-7 with the chemical composition given in Table 1 was characterised by the mechanical properties given in Table 2.

**Table 1.** Chemical composition of EN-GSJ-500-7.

| Chemical Composition, wt.% | | | | | | | | | |
|---|---|---|---|---|---|---|---|---|---|
| C | Si | Mn | P | S | Cr | Ni | Mg | Cu | Fe |
| 3.61 | 2.29 | 0.45 | 0.045 | 0.009 | 0.03 | 0.01 | 0.057 | 0.75 | rest |

**Table 2.** Mechanical properties of EN-GSJ-500-7.

| Tensile Strength, MPa | Conventional Yield Point, MPa | Elongation, % | Hardness, HB | Elastic Modulus, GPa |
|---|---|---|---|---|
| 500 | 340 | 7 | 220 | 169 |

The substrate samples had the dimensions of 100 mm × 15 mm × 5 mm. Prior to spraying, the surface of the substrates was sandblasted with loose corundum of 20 mesh granulation. The parameter of the substrate surface roughness Ra amounted to 30 μm. The technological parameters of the spraying process are given in Table 3. The average thickness of the applied coating was 200 μm.

**Table 3.** HVOF spraying parameters.

| Gun Movement Speed, mm/s | Oxygen, L/min | Kerosene, L/h | Powder Feed Rate g/min | Powder Feed Gas, L/min | Spray Distance, mm |
|---|---|---|---|---|---|
| 583 | 944 | 25.5 | 92 | nitrogen, 9.5 | 370 |

A light microscope (LM), a scanning electron microscope (SEM, Dual Beam Scios FEI, Eindhoven, Holand) and a transmission electron microscope (TEM, JOEL 2010 ARD, Croissy-sur-Seine, France) equipped with EDS spectrometers were used to study the microstructure and chemical composition of the coating/substrate type system. Coating/substrate type preparations for the transmission microscope in the form of a thin foil were obtained thanks to the use of ion thinning in a special device Gatan PIPS691V3.1 (Pleasanton, USA) for low-angle thinning [10]. The phase composition was determined by means of X-ray phase analysis on X'Pert Pro Panalytical diffractometer (Cambridge, UK) with CuK radiation (wavelength λ of 1.5418 Å). The measurement of the carbide coating porosity was carried out on microscopic photographs (LM) using Aphelion 3.0 programme to analyse the stereological parameters of the microstructure. The measurements of the indentation hardness were carried out for the precise assessment of the microhardness of the coatings. Measurements of indentation hardness ($H_{IT}$) and Young's modulus ($E_{IT}$) were carried out using a multifunction measuring platform equipped with a microhardness meter by Anton Paar. In this method, it is possible to evaluate the process of pressing an indenter into a material by measuring both force and displacement during plastic and elastic deformation. By registering the entire application and removal force cycle, hardness values equivalent to traditional hardness values as well as other material properties such as the pressing module can be determined. The advantage of this method is that all of the mentioned values can be calculated without having to measure the size of the imprint of an indenter. The hardness measurement with this method was carried out for a load force of 1 N and a load speed of 2 N/min. Five measurements were carried out for each sample. As part of the experiment, measurements of the surface roughness of coatings made by means of plasma spray using a confocal microscope were carried out. Three-dimensional images and their analysis allowed for a precise understanding of the geometrical structure of the examined surfaces.

Thermo-physical properties of the ductile cast iron and of the system coating/substrate were subjected to dilatometric analysis—there were determined values of dimensional changes ΔL/L of materials and thermal expansions coefficients in the solid state and as the function of a temperature, and there was measured the thermal conductivity using the most advanced sets of apparatus LFA 427 (Laser Flash Apparatus) produced by Netzsch (Selb, Germany). Measurements of changes in the thermal expansion were made using the high temperature dilatometer DIL 402C/4/G made by Netzsch under protective argon atmosphere in the range of temperature 100–700 °C at a heating rate 5 K/min. The LFA 427/4/G device of laser-flash type made by Netzsch was used to perform thermal conductivity measurements, using a pulse technique to determine the temperature conduction coefficient (**a**) in the range of temperature 20–700 °C, i.e., the laser-flash method of heating the front surface of the square-shaped sample with a short laser pulse, resulting in an increase of the temperature of the sample on its opposite surface measured as a function of time by means of the IR infrared detector. The measured signal allows to determine the temperature conductivity **a**, and to calculate the thermal conductivity **λ** based on the relationship:

$$\lambda(T) = a(T)cp(T)d(T) \tag{1}$$

where cp is proportionality factor called specific heat.

The strength of the coating/substrate joint was determined during a 3-point bending test on INSTRON 8800M strength machine using a specially designed holder for coating/substrate type samples with dimensions of 100 mm × 15 mm × 5 mm. The spacing of supports amounted to 70 mm, while the deformation rate was 1 mm/min. Three samples were used for a single test. Observations of the surface of fractures after a 3-point bending test were carried out using a scanning microscope.

Measurement of internal stresses in sprayed coatings was performer with X-ray non-destructive method (so called $sin^2y$). The test was carried out with an X-ray diffractometer, using monochromatic radiation of a lamp with a cobalt anode. The measurement of internal stresses was made at 4 points of a flat surface of the sample. Both the determination of measurement parameters and finding the position of diffraction lines at the assumed y-angles were carried out based on company programs

of APD or XRD Commander for phase analysis, equipped with the applied research apparatus. The obtained experimental inter-planar distances $d_{hkl}$ and X-ray elastic constants for the tested material constituted input data for the program calculating values of internal stresses. For the measurement of internal stresses in the carbide coating reflex (004) and elastic constants, Young's modulus 143 GPa and Poisson's ratio 0.29, were used.

The adhesion/cohesion bond strength of the thick plasma spray coatings by scratch testing on coating cross-sections (scratch bond strength test) were carried out using a multifunction measuring platform (Micro-Combi Tester, Buchs, Switzerland) equipped with Anton Paar scratch test heads according to the standard [11]. The tests were carried out on the cross-sectioned samples embedded in resin and then polished in a standard way as metallographic samples. The scratch test was done under the constant load and the indenter moved from the substrate through the coating into the resin where the sample was embedded. The test conditions are listed in Table 4.

**Table 4.** Scratch test conditions used in this study.

| Stylus | Scratch Mode | Load Applied | Scratch Length | Scratch Speed |
|---|---|---|---|---|
| Rockwell C, 100 µm | Constant load | 5, 10, 15 and 20 N | 1.2 mm | 0.4 mm/s |

After the test, the geometric values of the resulting one-shaped fracture was also measured: cone length Lx, with Ly and cone angle (image of the cone fracture area was taken immediately after scratching using on light microscope, shown in Figure 1). Among the Lx, Ly and cone angle values, the projected cone area, Acn = Lx.Ly was selected as the most characteristic factor because only Acn displayed a monotonic relationship to the scratching load.

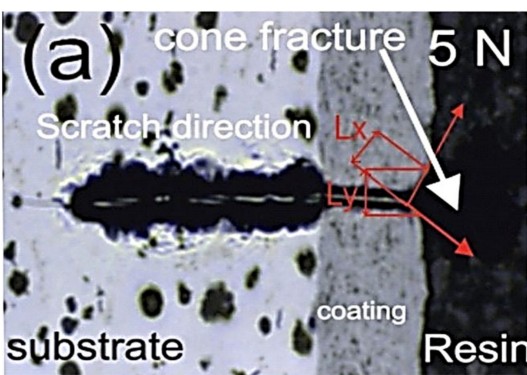

**Figure 1.** Schematics of the constant load scratch om cross sectioned sample.

Tests of abrasion resistance in abrasive slurry of ductile cast iron samples and coating systems of the type (1) $Cr_3C_2$-NiCr/ductile iron and (2) ($Cr_3C_2$-NiCr+Al)/ductile iron were carried out on the device for testing abrasion resistance of coatings and constructional materials. An abrasion test was carried out in an aqueous suspension of $Cr_3C_2$ powder (of average grain size less than 0.1 mm) at ambient temperature, with the following parameters: test time 3600 s, rotation speed 300 1/min, applied load 50 N. The friction node consisted of a stationary plate made of the tested material and a steel ball rotating at a set speed. The plate was pressed against the ball with a given force. The abrasive slurry was fed to the frictional contact zone. The registration of the test run took place using a computer with specialized software. The software enabled real-time calculation of load, path, rotation speed, temperature of the friction pair, depth of wear and rate of wear of both friction elements.

## 3. Results and Discussion

Selected results of observation of metallographic composite coatings ($Cr_3C_2$-NiCr+Al) sprayed with HVOF technique on a ductile cast iron substrate are shown in Figure 2. The coating has a

typical lamellar structure, i.e., arranged in layers flattened grains formed with powder particles that undergo sever deformation and geometric changes in the HVOF process. It is worth mentioning that during the spraying process the soft NiCr alloy, acting as a matrix, and Al particles, undergo strong deformation, while the ceramic particles remain not deformed, acting as particles strengthening the composite coating.

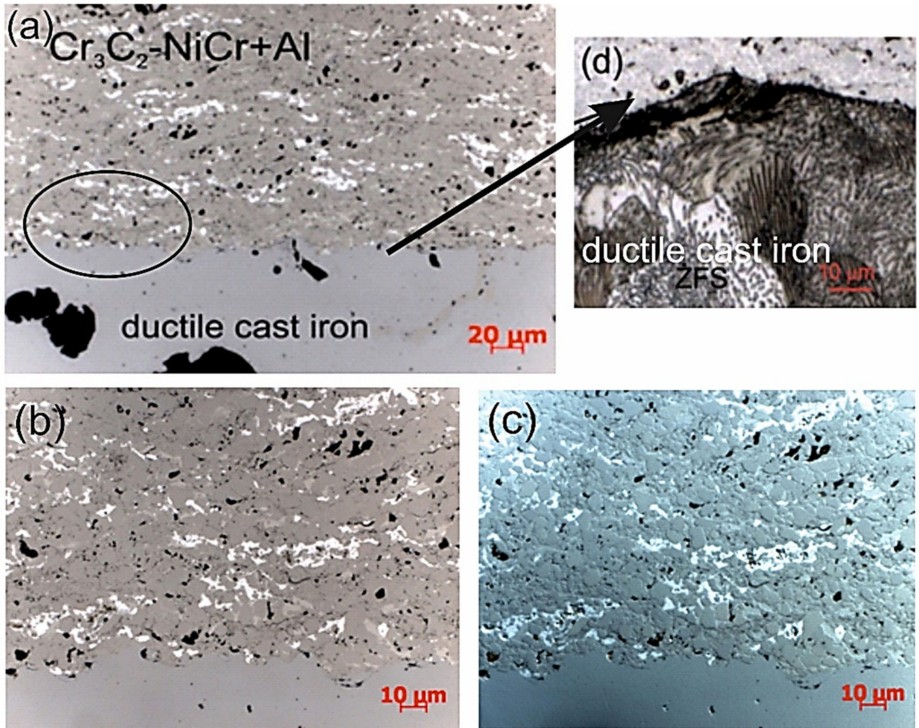

**Figure 2.** Microstructure of the composite coating ($Cr_3C_2$-NiCr+Al) deposited on ductile cast iron: (**a**) LM image; (**b**) magnified area selected in Figure 1; (**c**) details of the coating structure in differential interference contrast (DIC); and (**d**) cast iron structure composed of ferrite and perlite.

Hence, in the structure of the coatings, there are fine chromium carbide particles of various sizes, embedded in the NiCr alloy matrix and also band arranged Al particles, which upon hitting the substrate turn from spherical to elongated shape, reduce their height and extend parallel to the substrate surface. The Nomarski contrast shows details of the microstructure of coatings with bright melted Al particles, band arranged parallel to the boundary coating/substrate (Figure 2c). The coating is characterized by a compact construction with a low number of visible pores, without cracks and good adhesion to the substrate (the boundary between the substrate and the coating is continuous), which indicates favourable conditions of the application process, ensuring adequate adhesion of the coating to the substrate. In the cast iron structure at the coating/substrate interface from the substrate side, no changes were observed after the spraying process (initially and after the spraying the cast iron matrix is composed of ferrite and perlite—Figure 2d). The average porosity of the $Cr_3C_2$-NiCr + Al composite coating is 2.4%. For a coating without Al particles the porosity does not exceed 5%. This low porosity value due to the high impact velocity of the coating particles resulted in high density and high cohesive strength of individual splats [12]. Moreover, the addition of Al is beneficial to reducing the porosity of the coating, since Al particles as compared to $Cr_3C_2$ particles have a much lower melting point and better fill pores in the coating. It is worth noting also, that the composite coating has a relatively low surface roughness, the value of the Ra roughness parameter is 4.2 μm. For a coating without Al particles, the value of this parameter is 6.6 μm. The addition of Al is beneficial to reducing the porosity of the coating, since Al particles as compared to WC particles have a much lower melting point and better fill pores in the coating

SEM observation of the cross section of the composite coating ($Cr_3C_2$-NiCr+Al) revealed evenly spaced $Cr_3C_2$ particles in the NiCr matrix and finely divided to approximately 1 μm and also even distribution of molten aluminium particles (Figure 3). There are visible dark grey grains of the phase with a high chromium content (the EDS spectrum of point 1), lighter areas are the zones rich in nickel with a small amount of chromium (the EDS spectrum of point 2), and black fields are the areas of the occurrence of Al phase (the EDS spectrum of point 3).

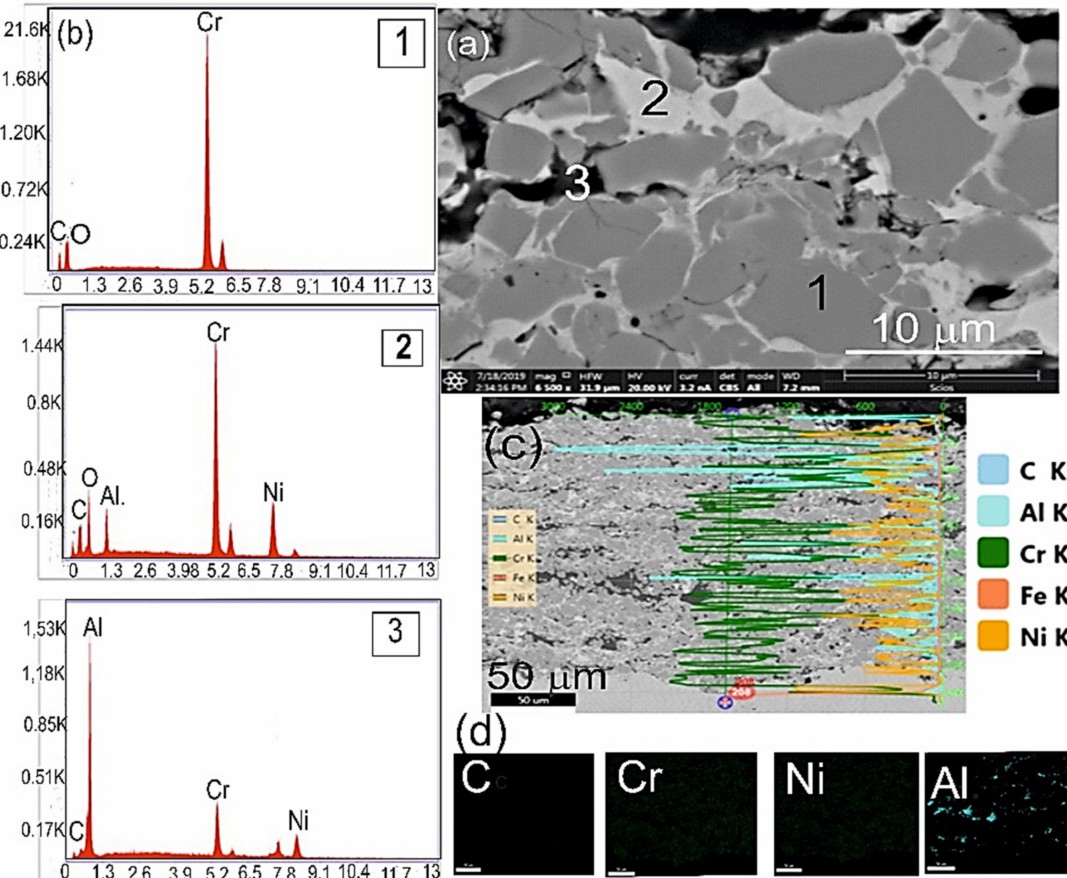

**Figure 3.** Scanning micrographs of the composite coating ($Cr_3C_2$-NiCr+Al) deposited on ductile cast iron: (**a**) SEM image; with (**b**) EDS spectra taken from the marked points: 1, 2 and 3; (**c**) linear distribution of concentrations of C, Al, Cr, Fe and Ni; and (**d**) map of distribution of concentrations of C, Cr, Ni, Al taken from the region of interface.

The results of indentation measurements, i.e., hardness $H_{IT}$ and Young's modulus $E_{IT}$ of the tested coatings, are from 14.16 to 15.57 GPa and 301.28 to 301.88 GPa, respectively, for the areas of $Cr_3C_2$ occurrence. On the other hand, for the areas of the occurrence of Al particles, the values of hardness and Young's modulus are from 6.57 to 8.73 GPa and from 162.43 to 224.91 GPa respectively. Measurements of microhardness of coatings show significant dispersion, which is associated with their microstructure. Doping with metallic particles resulted in local reduction in the hardness of the coating, which consequently reduces its brittleness. It is worth noting here that after spraying the composite coating on ductile cast iron, a 6-fold increase in ductile iron hardness is observed compared to the initial state, i.e., without the coating (metal matrix hardness equals 2.53 GPa). The coating hardness increase is the result of a deformation effect of the particles falling on the substrate during the spraying process and is also due to the presence of very hard and fine ceramic particles. The joining area of the coating and substrate shows a high degree of the substrate deformation by the falling powder particles forming the coating, which significantly improves the bonding between the coating and the substrate.

Detailed microstructure examination of the coating performed on a thin TEM film from the sample cross-section showed nanocrystalline band-like structure. In the microstructure of coating, there are longitudinal bands 50–150 nm thick, arranged parallel to each other (Figure 4), inside which there are nanocrystalline $Cr_3C_2$ particles. Electron diffractions ring pattern confirmed the nanocrystalline nature of the coating structure. Thanks to the EDS (Energy Dispersive X-ray Spectroscopy) technique, a point analysis of the chemical composition in the coating was obtained and the elements constituting the coating—Cr, Ni and Al—were identified.

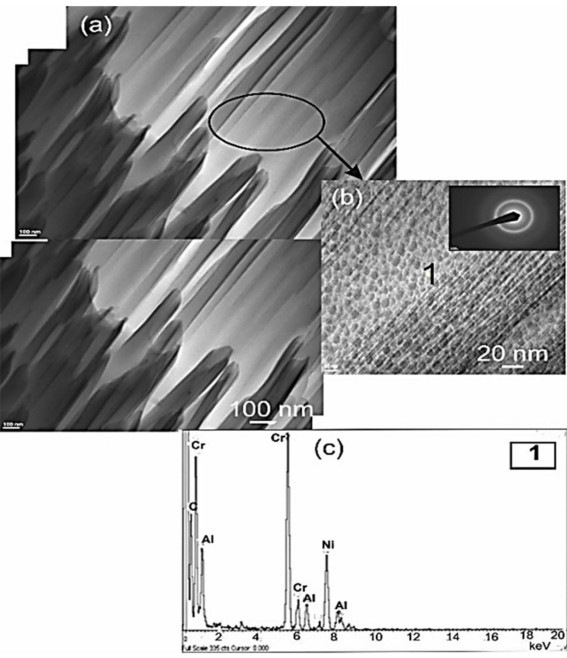

**Figure 4.** TEM analysis of the composite coating ($Cr_3C_2$-NiCr+Al) deposited on ductile cast iron: (**a**) TEM image with representative (**b**) area diffraction pattern that indicates the formation of nanocrystalline structure and with (**c**) EDS spectrum taken from the marked point.

Examination by X-ray phase analysis of the composite coating shows that in the composition can be identified $Cr_3C_2$ chromium carbide and NiCr matrix material and the Al phase. Phase analysis performed on the basis of diffraction studies, except for the occurrence of the ceramic phase $Cr_3C_2$ and metallic phases NiCr and Al, did not show any new phases formed during spraying (Figure 5), i.e., carbides $Cr_7C_3$ and $Cr_{23}C_6$, formed in the decarbonisation process of $Cr_3C_2$ carbide [13]. X-ray examinations do not confirm the occurrence of these carbide phases because the main diffraction peaks $Cr_7C_3$ and $Cr_{23}C_6$ coincide with the lines corresponding to the phases NiCr and $Cr_3C_2$ [14]. In addition, the diffraction lines of the revealed phases are not widened, which indicates a low degree of elastic-plastic deformation of the metallic phases, NiCr and Al. Additionally, there were determined volumetric shares and average crystallite sizes of individual phases in the tested coating (Table 5).

**Table 5.** Detailed results of XRD analysis.

| Composition | Weight Percentage of Phase Composition, % | Crystal Size from XRD $D_{XRD}$, nm |
|:---:|:---:|:---:|
| $Cr_3C_2$ | 73.8 | 50 |
| NiCr | 7.8 | 30 |
| Al | 18.5 | 14 |

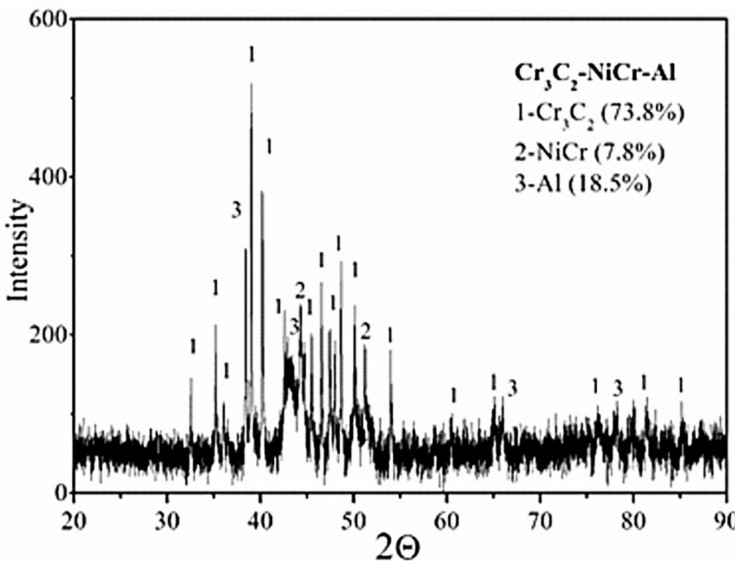

**Figure 5.** X-ray diffraction pattern of the composite coating (Cr$_3$C$_2$-NiCr+Al) deposited on ductile cast iron by HVOF.

The content of Cr$_3$C$_2$ was 73.8 wt % , and the contents of NiCr and Al phase were, respectively, 18.5 wt % and 7.8 wt %. It is worth mentioning that the values of the average crystallite sizes of the individual phases demonstrate the nanocrystalline nature of the coating. The results of dilatometric measurements—dimensional changes ∆L/L and coefficients of thermal expansion in the solid state as a function of temperature—are presented for ductile cast iron and coating system in the figures (Figure 6). The greatest thermal expansion in the whole range of temperature is characterized by the ductile cast iron. Up to a temperature of approximately 350 °C, the similar type of thermal expansion is shown by *Cr$_3$C$_2$-NiCr*/ductile cast iron and *(Cr$_3$C$_2$-NiCr+Al)*/ductile cast iron systems. Above this temperature, the expansion of the coating systems is less than the expansion of ductile cast iron (they have higher thermal stability at high temperature). In addition, above this temperature, the composite coating (Cr$_3$C$_2$-NiCr+Al) applied to the ductile cast iron has a slightly higher thermal expansion compared to the Cr$_3$C$_2$-NiCr coating. The figures (Figure 7) show diagrams of dependence of the temperature and thermal coefficient as a function of the temperature for ductile cast iron and coating systems, and the Table 6 presents a comparison of temperature and thermal conductivity values of the tested materials.

**Table 6.** Temperature and thermal conductivity coefficients in function of temperature for ductile cast iron and systems: (Cr$_3$C$_2$-NiCr+Al)/ductile cast iron and (Cr$_3$C$_2$-NiCr)/ductile cast iron.

| Temperature, °C | 100 | 200 | 300 | 400 | 500 | 600 | 700 |
|---|---|---|---|---|---|---|---|
| **Substrate** | **Ductile Cast Iron** | | | | | | |
| Temperature conductivity, mm$^2$/s | 8.13 | 7.96 | 7.58 | 7.06 | 6.42 | 5.61 | 4.32 |
| Thermal conductivity, W/mK | 28.88 | 28.45 | 25.59 | 21.96 | 18.66 | 18.22 | 18.08 |
| **Coating** | **Cr$_3$C$_2$ -NiCr+ Al** | | | | | | |
| Temperature conductivity, mm$^2$/s | 8.00 | 7.90 | 7.59 | 7.17 | 6.52 | 5.69 | 4.59 |
| Thermal conductivity, W/mK | 28.41 | 28.25 | 25.62 | 22.29 | 18.96 | 18.47 | 19.24 |
| **Coating** | **Cr$_3$C$_2$ -NiCr** | | | | | | |
| Temperature conductivity, mm$^2$/s | 8.02 | 7.96 | 7.64 | 7.17 | 6.58 | 5.80 | 4.56 |
| Thermal conductivity, W/mK | 28.49 | 28.46 | 25.77 | 22.28 | 19.13 | 18.86 | 19.12 |

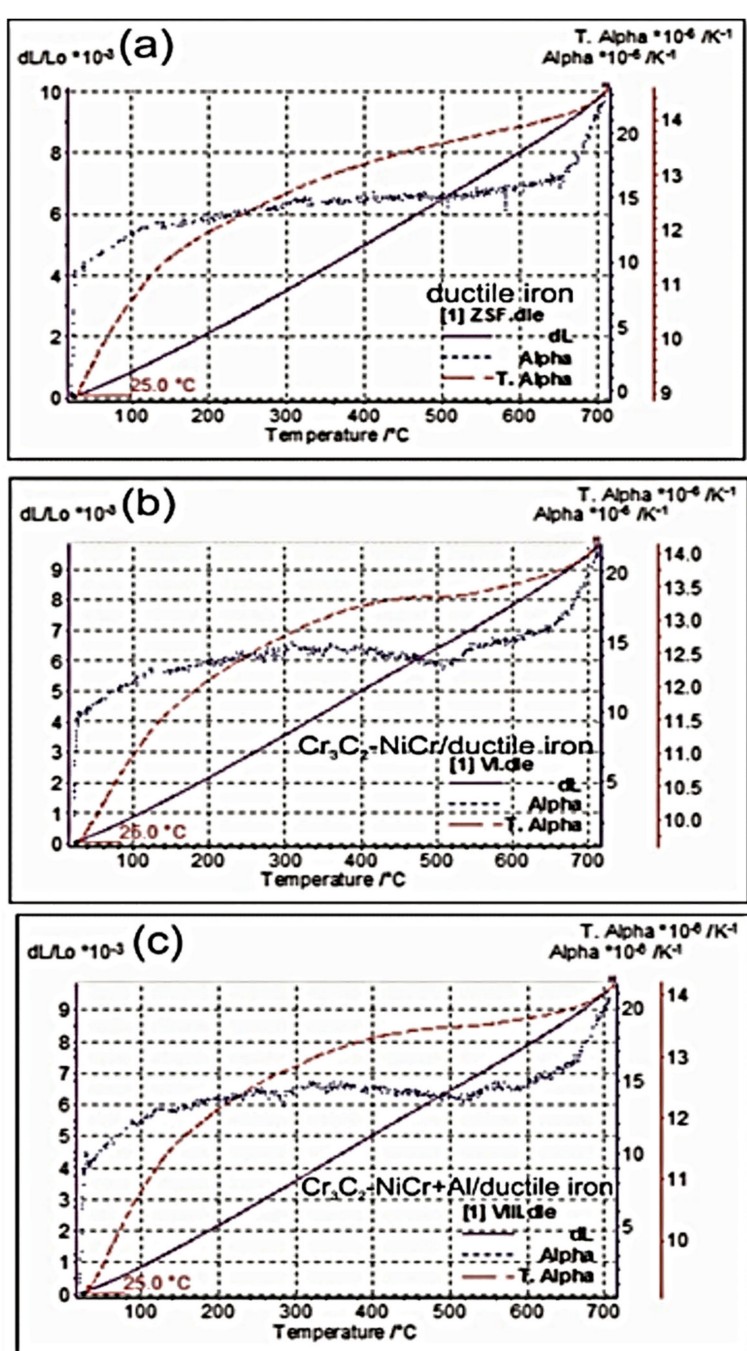

**Figure 6.** Plotted relationships between the thermal expansion (dL), the linear thermal expansion coefficient (Alpha), the average thermal expansion coefficient (T.Alpha) and temperature for: (**a**) ductile cast iron; (**b**) (Cr$_3$C$_2$-NiCr)/ductile cast iron; and (**c**) (Cr$_3$C$_2$-NiCr+Al)/ductile cast iron.

The analysis of dependences a(T) and λ(T) listed in Table 6 and diagrams allow to state that the coating system in relation to the ductile cast iron is characterized by better temperature and thermal conductivity in the range of temperature 300–700 °C. A slight difference in thermal deformations (difference in coefficients of thermal expansion of the coating and the substrate) suggests that the spraying process generates internal stresses related to the process of the coating formation and building up (rather than thermal). Hence, the differences in thermal conductivity of the coating and the substrate do not affect the state of stress.

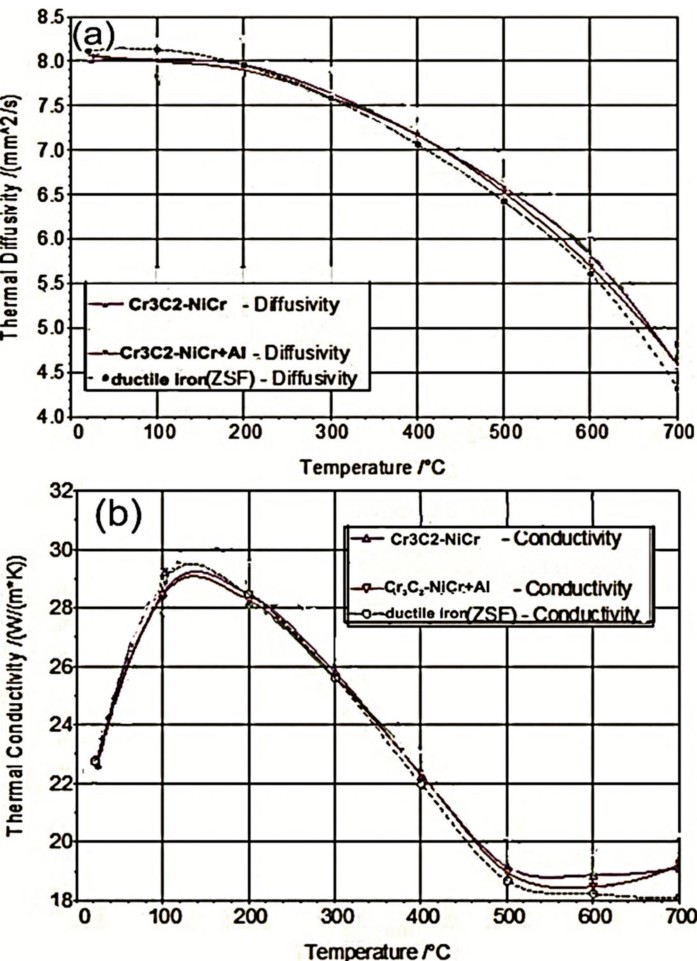

**Figure 7.** Plotted relationship between temperature conductivity a(T) (**a**) and (**b**) thermal conductivity λ(T) for: (Cr$_3$C$_2$-NiCr)/ductile cast iron, (Cr$_3$C$_2$-NiCr+Al)/ductile cast iron and ductile cast iron.

The results of tests calculations of main stresses (σ$_1$, σ$_2$) in the produced coatings Cr$_3$C$_2$-NiCr and (Cr$_3$C$_2$-NiCr+Al) and the orientation of the main stress σ$_1$ of the tested sample are listed in Table 7. In the tested coatings Cr$_3$C$_2$-NiCr and (Cr$_3$C$_2$-NiCr+Al) sprayed onto a ductile cast iron substrate, tensile stresses occur. However, in (Cr$_3$C$_2$-NiCr+Al) coatings there is a tendency to lower the level of tensile stresses (oriented at an angle of 40° ± 30° clockwise from the direction marked on the sample) by approximately 14% to 18%. There is no strong defecting in the microstructure of the formed coatings after the HVOF spray process, which is confirmed by small deformation of the material and the formation of tensile stresses. Tensile stresses, and in particular their considerable concentration, in the area of the coating-substrate interface cause the cracks in the coatings, and even delamination, which results in a significant reduction of durability of the coating. Internal stresses in the sprayed coatings are the sum of stresses from the cooling of liquid drops of the coating material and stresses during cooling of the formed coating and the substrate as a whole [15]. Their value is influenced by such factors as differences in the material of the coating and the substrate in terms of chemical composition, phase composition, microstructure, texture and defects density and, in addition, the high temperature of the spraying process. In particular, reducing the difference in the coefficients of thermal expansion of the coating and the substrate reduces the level of internal stress (which are undoubtedly one of the reasons for initiating micro-cracks and macro-cracks in the coating during exploitation), which in turn increases the mechanical durability of the coating.

**Table 7.** Results of analysis of residual stresses in $Cr_3C_2$-NiCr and ($Cr_3C_2$-NiCr+Al) coatings.

| Description | $Cr_3C_2$-NiCr | $Cr_3C_2$-NiCr+Al |
|---|---|---|
| Internal stress $\sigma_1$ [MPa] | 420 ± 60 | 360 ± 90 |
| Internal stress $\sigma_2$ [MPa] | 232 ± 70 | 190 ± 100 |
| Orientation of the main stress $\sigma_1$ (clockwise from the direction marked on the sample) | 40° ± 30° | 40° ± 30° |

The Figure 8 shows a comparison of bending test results for the system: $Cr_3C_2$-NiCr/ductile cast iron and ($Cr_3C_2$-NiCr+Al)/ductile cast iron in the relation of bending stress-deflection. Values of maximum bending stresses for $Cr_3C_2$-NiCr/ductile iron and ($Cr_3C_2$-NiCr+Al)/ductile iron systems are 534 ± 7 MPa and 567 ± 10 MPa, respectively. In the tested systems, the bending curves have parabolic characteristic. However, for the ($Cr_3C_2$-NiCr+Al)/ductile cast iron system on the bending curve, there is a longer range of the deflection pathway, during which the stress gently increases and then drops.

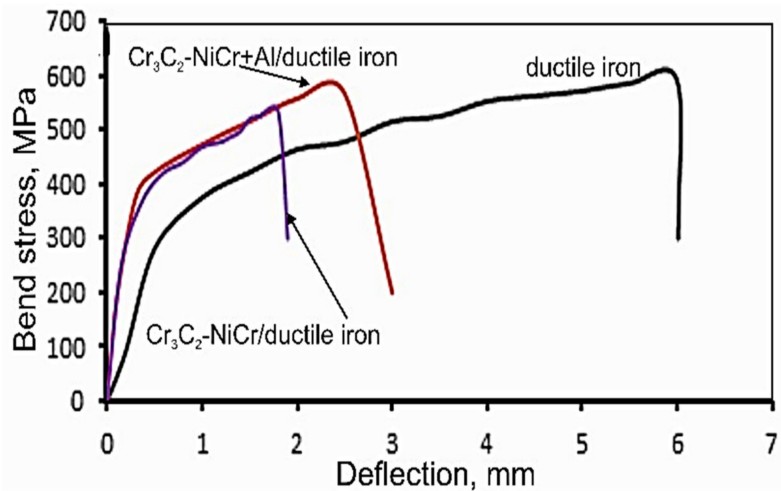

**Figure 8.** Bend test curves recorded for systems: ($Cr_3C_2$-NiCr)/ductile cast iron and ($Cr_3C_2$-NiCr+Al)/ductile cast iron.

The value of the deflection, followed by a drop in stress leading to the destruction of the sample, is 2.5 mm. However, for a coating system without Al particles, there is no such long range of the deflection path. Comparing the obtained curves, it can be concluded that for the $Cr_3C_2$-NiCr/ductile cast iron system, the strength parameters of the bending process are reduced, and the deflection is reduced to 1.8 mm. It is worth noting that in a coating system without Al particles, the coating is less plastic, which consequently limits the dissipation of plastic deformation energy and the intensively increasing load that causes crack propagation and a smaller deflection range.

Observation of sample fracture after the bending test made on a scanning microscope (Figure 9) indicates that in the $Cr_3C_2$-NiCr/ductile cast iron system, the destruction takes place both in the coating near the boundary of coating/substrate interface and along the boundary of interface, while in the ($Cr_3C_2$-NiCr+Al)/ductile cast iron system the destruction takes place only along the boundary of interface. When spraying ceramic material on a metal substrate, there are differences in thermal-physical and mechanical properties of both materials, which generate a certain state of the internal stresses in the coating/substrate system. The size and distribution of these stresses affect the mechanical durability of the coating-substrate join area, in particular the adhesion of the coating to the substrate, its scratch resistance and microhardness [16].

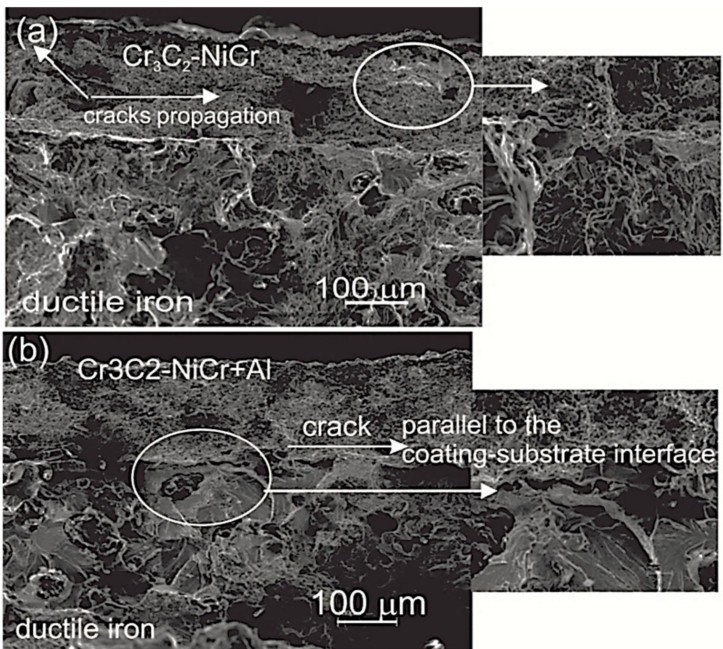

**Figure 9.** Scanning micrographs of the fracture surface of the systems: (**a**) ($Cr_3C_2$-NiCr)/ductile cast iron; and (**b**) ($Cr_3C_2$-NiCr+Al)/ductile cast iron after bend test.

The results of the scratch bond strength test performed on the cross-section of the coating/substrate system at a constant load of 5, 10, 15 and 20 N are listed in the Table 8. Generally, for the system with the $Cr_3C_2$-NiCr coating, the calculated values Acn (projected as cone) in the whole load range are higher than that on the composite coating system, which consequently indicates that the composite coating system has a higher scratch bond strength. For both coating systems, cone shaped fracture occurs inside the coating, indicating cohesive failure in the coating/substrate system (Figure 10). However, at the maximum load (20 N) there appear in the coating $Cr_3C_2$-NiCr larger cracks around the scratch, even leading to the delamination of the coating from the substrate (adhesive failure). An introduction of Al particles to the coating material increases the plasticity of the coating and scratch bond strength of this coating.

**Table 8.** Averaged scratch bond test results of the investigated coatings.

| Coating /Load | 5 N | | | 10 N | | | 15 N | | | 20 N | | |
|---|---|---|---|---|---|---|---|---|---|---|---|---|
| | Lx μm | Ly μm | $A_{cn}$ $\times 10^{-3}$ (mm²) | Lx μm | Ly μm | $A_{cn}$ $\times 10^{-3}$ (mm²) | Lx μm | Ly μm | $A_{cn}$ $\times 10^{-3}$ (mm²) | Lx μm | Ly μm | $A_{cn}$ $\times 10^{-3}$ (mm²) |
| $Cr_3C_2$-NiCr+Al | 30 | 46.77 | 1.41 | 34.33 | 56.82 | 1.95 | 37.42 | 94.36 | 3.53 | 63.85 | 133 | 8.49 |
| $Cr_3C_2$-NiCr | 38.42 | 56.65 | 2.18 | 46.35 | 58.95 | 2.73 | 43.83 | 112 | 5.48 | 72,26 | 140 | 10.12 |

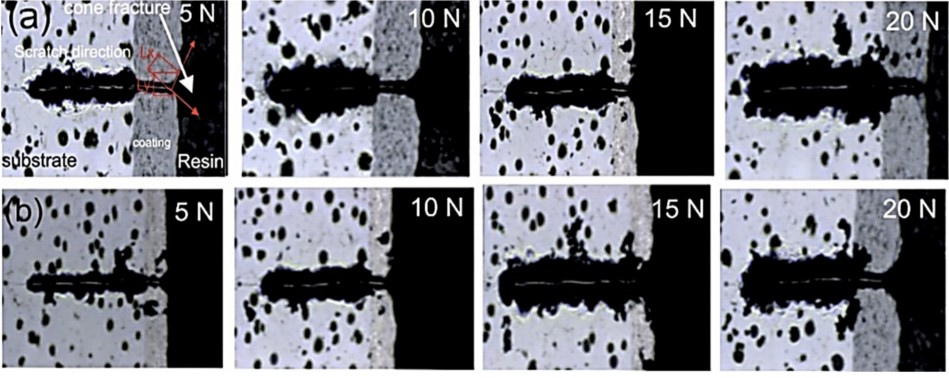

**Figure 10.** Cone-shaped fracture occurring during the scratch bond strength test for coatings: (**a**) $Cr_3C_2$-NiCr+Al; and (**b**) $Cr_3C_2$-NiCr.

The abrasive wear of the coatings of the type: $Cr_3C_2$-NiCr, ($Cr_3C_2$-NiCr+Al) and ductile cast iron was evaluated by the depth of wear path (L) and the rate of wear (Vz—as the ratio of the depth of the wear path and the test route). The measurement results presented in the figures (Figure 11) indicate that coating systems: $Cr_3C_2$-NiCr/ductile cast iron and ($Cr_3C_2$-NiCr+Al)/ductile cast iron show significant lower wear in relation to the ductile cast iron (expressed by the depth of wear).

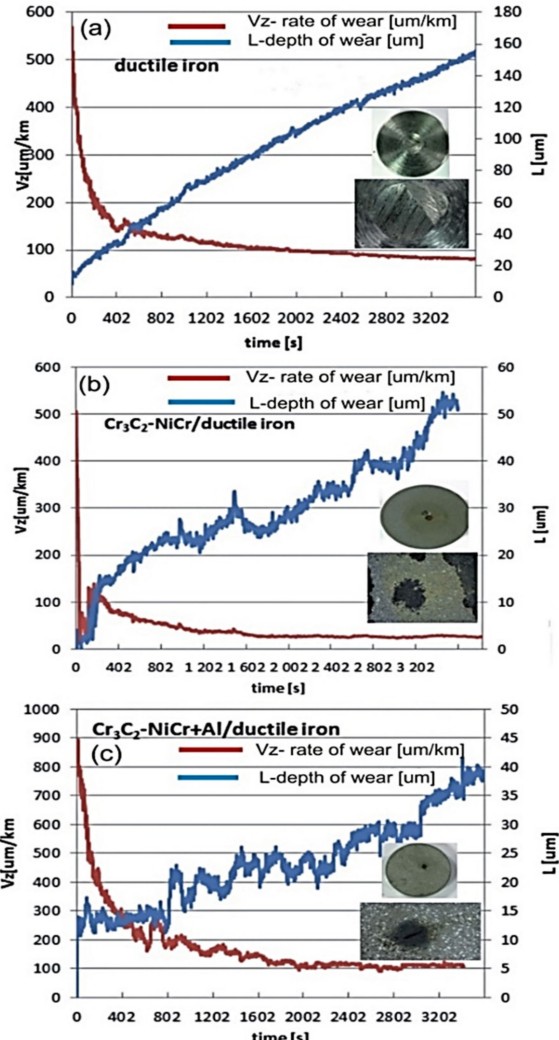

**Figure 11.** The results of the wear resistance tests of: (**a**) ductile cast iron; (**b**) ($Cr_3C_2$-NiCr)/ductile cast iron; and (**c**) ($Cr_3C_2$-NiCr+Al)/ductile cast iron.

In addition, the measurement results indicate that the coating with an addition of Al (ceramic-metallic) has a much better abrasion resistance than the $Cr_3C_2$-NiCr coatings. The better wear resistance of the composite coating is related to its higher mechanical durability. In addition, the lower porosity of the composite coating makes it difficult to penetrate the surface structure by the abrasive slurry particles. The morphology of the wear area of the composite coating is shown in Figure 12. Studies of the morphology of the wear area of the composite coating revealed presence lips, craters, cracks and micropores. It is worth mentioning that the worn surface is relatively smooth; microcutting occurs in the area of metallic phases, a soft nickel-chromium matrix NiCr and molten Al particles, which consequently leads to difficulties in revealing and removing of carbide particles [15–18]. Mechanisms responsible for the degradation of the coating under erosive wear can be divided into two types: plastic and brittle. Elements such as cutting and craters surrounded by lips indicate the plastic nature of wear, while cracks and chipping indicate the fragile nature of wear [19–21]. Based on the

above results, it can be suggested and emphasized, that on the interface of coating/substrate there were found no defects that could reduce wear resistance or that could cause poor adhesion of the coating to the substrate. The high wear resistance of the composite coating ($Cr_3C_2$-NiCr+Al) is influenced by both a strong bonding of the $Cr_3C_2$ particles with NiCr binder phase, as well as a strong bonding of the Al molten particles with the $Cr_3C_2$ coating material, consequently hindering material removal during the erosion test.

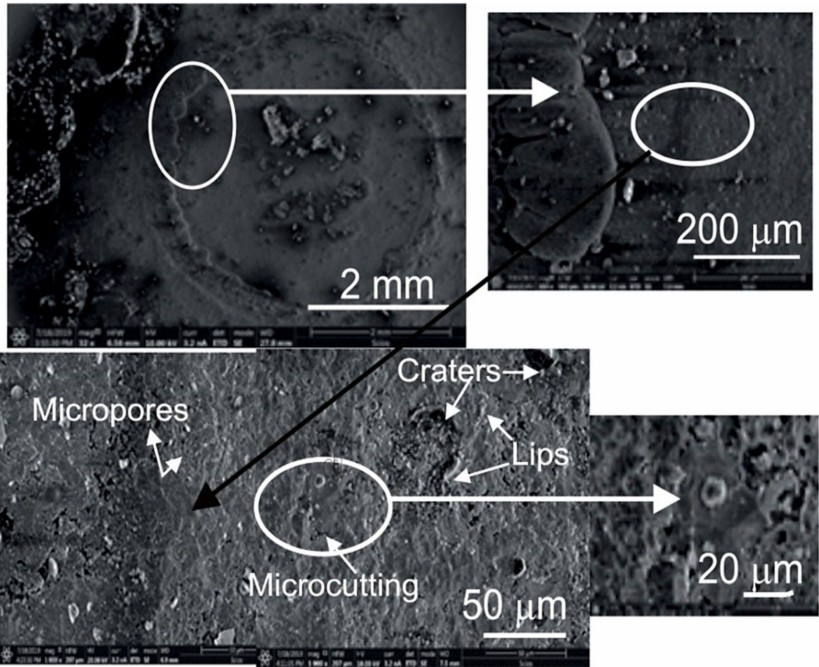

**Figure 12.** The morphology of the wear area of the composite coating ($Cr_3C_2$-NiCr+Al).

## 4. Conclusions

Based on the tests and analysis of results performed, the following conclusions were drawn:

- The composite coating ($Cr_3C_2$-NiCr+Al) applied by the HVOF method on ductile cast iron is characterised by low porosity, compact structure, good adhesion to the substrate and high hardness. The coating microstructure contains molten Al particles and very fine $Cr_3C_2$ particles embedded in a nickel-chromium alloy matrix, reaching nanocrystalline dimensions.

- The composite ($Cr_3C_2$-NiCr+Al) coating structure provides good resistance to cracking. The damage occurs along the coating/substrate interface. The cracks initiated near the interface do not violate the interface and do not break into the crack in the substrate. In addition, lower values of both the residual stress of the ($Cr_3C_2$-NiCr+Al) coating surface as well as microhardness leads to an increase in bend strength of the composite coating-substrate system.

- The system ($Cr_3C_2$-NiCr+Al)/ductile cast iron is characterized by higher values of the bond strength (measured by scratch bond strength test) than the system $Cr_3C_2$-NiCr/ductile cast iron, which is associated with better quality and ductility of the coating.

- Carbide coatings made with the HVOF method have good resistance to erosive wear. However, the ($Cr_3C_2$-NiCr+Al) composite coating on the ductile cast iron has better (almost 20%) wear erosion resistance than the $Cr_3C_2$-NiCr coating, which is related to the effect of plasticizing the coating by addition of metallic particles to the base ceramic powder. The surface morphology after the erosion test indicates the combined mode of erosion mechanism (i.e., ductile and brittle).

**Author Contributions:** M.K. conceived and designed the experiments, also wrote the paper. L.B. and A.T. performed the experiments and analysed the data.

**Funding:** This research received no external founding.

**Acknowledgments:** The work was financed from the subsidy of the AGH Non-Ferrous Metals Faculty (contract No. 16.16.180.006).

**Conflicts of Interest:** The authors declare no conflict of interest.

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
