# Peer review of "Microstructure, Mechanical Properties and Wear Behavior of High-Velocity Oxygen-Fuel (HVOF) Sprayed (Cr3C2-NiCr+Al) Composite Coating on Ductile Cast Iron"

_coatings, doi:10.3390/coatings9120840_

Round 1
Reviewer 1 Report
1. The paper should not be accepted for publication in its present form.
2. The author used a lot of self-made terms, which is hard to catch.
3. The author used many ambiguous pictures, which is hard for readers to understand the purpose expressed by the author.
4. The reviewer strongly doubt that the Al additive can improve the anti-abrasive properties of NiCr-Cr2C3. How the author compared the anti-abrasive performance on three different coatings.
5. It is strongly uncertain that the EDS spectrum can tell the diffusion phenomenon in Figure 2, and it is doubtful that no new phase is formed since diffusion occurred.
6. More details of the powder should be provided, and the phase transition from crystal to amorphous in spraying process should be illustrated.
7. Figure 4 and Table 4 indicate that the coating is 100% crystal structure, while in TEM there is amorphous phase?
Author Response
Thank you very much for the valuable comments posted in the review.
1. The text has been corrected for:
- technical terms,
- description of the figures and their improvement,
- entries deleted regarding diffusion in the area of the coating-substrate
interface , because the micrograph does not show the interface, and the
amorphous nature of the matrix structure,
- no morphological studies were performed at work of particles in the starting powders, and no quantitative and qualitative phase analysis of the powders was performed.
Commercial powder was used in the work with approval (Dimalloy).
- improving the methodology.
2. The following article is attached with marked amendments

Reviewer 2 Report
The authors have done a good work that could provide valuable information in this field.
However, some issues should be resolved prior to next steps.
1. Line 12, “…in an amount…” is not necessary here and can be deleted here. Please indicate the percentage “10%” is weight “wt.” or something else. Please check throughout the manuscript for the same issues and fix them all.
2. Line 15 to 16, “…of type:…” is not necessary and should be deleted here. Please check throughout the manuscript for the same issues and fix them all.
3. Line 19, please check “…splat by splat…” and fix it.
4. Line 33 to 34, add “of” between “method” and “HVOF”. Please check throughout the manuscript for the same issues and fix them all.
5. Line 34, the colon “:” should be deleted after “…as…”.
6. Line 38, the colon “:” can be replaced with “such as” in this case.
7. Line 45, “grow” should be changed to “growth” in this case.
8. Line 45 to 46, please check what is “finely divided grain”?
9. Line 46, please check the use of “both” it could be deleted here.
10. Line 61, please check “in terms of”, it could be not necessary here and can be deleted.
11. In Table 2, your experimental results of Young’s modules can be introduced.
12. Line 85, please check “…20÷90°…”.
13. Line 85, the wavelength can be provided for CuK radiation.
14. Line 107, 111 to 112, 116, please check the concept of “temperature conductivity” that does not make sense. Did you intend to say “heat conductivity” or “thermal conductivity” here? Please check throughout the manuscript for the same issues and fix them all.
15. Line 107, “by” can be changed to “using”.
16. Line 108, please explain what is “LFA” here.
17. Line 117, the equation may need a label (number).
18. Line 123, "...performer by..." should be changed to "... performed with..." in this case.
19. Line 124, "by" should be changed to "with" in this case.
20. Line 131, what is assumed elastic constant? and why assumed? did you intend to say elastic modules?
21. Line 131, the colon ":" is not properly used here and can be deleted.
22. Line 133, 325, "scratch bond strength" does not make sense. Please check and correct it. Please check throughout the manuscript for the same issues and fix them all.
23. Line 136, add an article "a" or "the" before "constant".
24. Line 139 to 140, the parameters can be summarized in a table.
25. Line 143, please check why Fig 9 is mentioned first in this manuscript? Also, both "Fig. x" and "Figure x" are used throughout this manuscript, Please check throughout the manuscript to make them in consistent format so that it is easy for the readers to search.
26. Line 152, please check "at a set force".
27. Line 153, add an article "a" before "computer".
28. Line 154, please check "The software enabled to:..." for grammatical issues and resolved them.
29. Line 156, "1...." should be changed to "3....".
30. Line 159, "by" should be changed to "with" in this case.
31. Line 165, Figure 1 needs a general caption and then followed with sub-captions "(a)...(b)...".
32. Line 170 to 171, please check the sentence for grammatical issues and fix them.
33. Line 171, "structure" should be changed to "microstructure" here.
34. Line 176, please check "at the boundary interface" for grammatical issues and fix them.
35. Line 181, "results" should be changed to "resulted" in this case.
36. Line 183 to 184, please check the sentence for grammatical issues and fix them.
37. Line 193, "...and vice versa was found..." should be changed to "...was found, and vice versa...".
38. Line 196, Figure 2 needs a general caption and then followed with sub-captions "(a)...(b)...".
39. Line 200, "structure" should be changed to "microstructures" here.
40. Line 208, add "the" before "falling".
41. Line 209, "connection" can be changed to "bonding" in this case.
42. Line 212, add "the" before "areas".
43. Line 214, what is "banding nature"?
44. Line 217, what is "patters"?
45. Line 219, please check "infer about", "about" seems not necessary here.
46. Line 220, please check "there was obtained" for grammatical issues and fix them.
47. Line 223, Figure 3 needs a general caption and then followed with sub-captions "(a)...(b)...".
48. Line 241, add "analysis" after "XRD".
49. Line 242, "xx% in weight" can be written as "xx wt.%". Please check throughout the manuscript for the same issues and fix them all.
50. Line 247, "drawings" should be changed to "figures" in this case.
51. Line 247, "highest" can be changed to "greatest" in this case.
52. Line 249, please check "...is shown by systems of the type..." for grammatical issues and fix them.
53. Line 250, add "the" before "coating".
54. Line 255, please check "...and in Table 5 are presented numerical values." for grammatical issues and fix them.
55. Line 266, please check "...pressure state is not affected by the calculated ..." for logical issues and fix them.
56. Line 269, Figure 6 needs a general caption and then followed with sub-captions "(a)...(b)...".
57. Line 271 to 272, please check the fonts.
58. Line 273, what is "...test calculations..."?
59. Line 273, what is "...coating of type..."?
60. Line 281, please check"...boundary coating-substrate interface..." for logical and grammatical issues and fix them.
61. Line 281, "...formation of..." is not necessary here.
62. Line 285, "value" should be plural here.
63. Line 285, ":" should be deleted here.
64. Line 290, please check for grammatical issues and fix them.
65. Line 296, "534 MPa±7 and 567 MPa±10" should be written as "534±7 MPa and 567±10 MPa", respectively. Please check throughout the manuscript for the same issues and fix them all.
66. Line 297, "character" should be changed to "characteristic" in this case.
67. Line 308, add "that" before "cause".
68. Line 324, "for" should be changed to "...that on...".
69. Line 330, "increases" is not necessary here.
70. Line 337, please check "...there were performed..." for grammatical issues and fix them.
71. Line 346, please check "...can be observed lips..." for grammatical issues and fix them.
72. Line 347, "surface of wear" can be changed to "worn surface".
73. Line 353, "..., that on the boundary..." can be changed to "...on the interface of...".
74. Line 358, please check "...makes them difficult to remove..." for grammatical errors and correct them.
75. Line 365, "1...." should be changed to "4....".
76. Line 366 to 390, the writing check throughout the Conclusions contains many grammatical issues. Please check and fix them all.
77. Line 393, the title of Reference should be not numbered. Please correct and renumber all the references in the list of References and in the text to make sure the citations of references are all correct.
78. Line 443 and 445, why there are two numbers without references?
Author Response
Thank you very much for the valuable comments posted in the review.
1. The text has been corrected for:
- only results regarding hardness measured by the indentation method are provided - for simplicity. Microhardness measurements by the Vikers method (conventional) have been given earlier because coincided with the hardness values obtained by the indentation method. These values were also removed from Fig. 2
- the text and Fig. 11 provide an explanation of the abrasion test parameters: L - depth of wear and Vz - wear factor as a ratio of the depth of wear and the road
- improving the methodology and discussion.
Below is the text with the corrected elements marked.

Reviewer 3 Report
The advantage of Cr2C3-NiCr with Al coating looks clear, but needs to be revised a minor thing.
- There is no experimental procedure how the Cr3C2-NiCr-Al were prepared and sprayed in detail.
- There is no explanation on 1388HV0.1, 87Hv0.002 and 1297 Hv0.1 in Fig. 1(b). It looks they explained it in Fig. 2.
- In Fig. 10, explain what Vz and L means exactly.
- 1. Result and discussion --> 3. Results and discussion.
Author Response
Thank you very much for the valuable comments posted in the review.
1. The text has been corrected for:
- only results regarding hardness measured by the indentation method are provided - for simplicity. Microhardness measurements by the Vikers method (conventional) have been given earlier because coincided with the hardness values obtained by the indentation method. These values were also removed from Fig. 2
- the text and Fig. 11 provide an explanation of the abrasion test parameters: L - depth of wear and Vz - wear factor as a ratio of the depth of wear and the road
- improving the methodology and discussion.
2. Below is the text with the corrected elements marked.

Round 2
Reviewer 1 Report
The manuscript was revised extensively and can be accepted for publication.
Reviewer 2 Report
1. Line 64, please indicate the percentage “25%” is weight “wt.” or something else. Please check throughout the manuscript for the same issues and fix them all.
2. Line 145, both "Fig. x" and "Figure x" are used throughout this manuscript, Please check throughout the manuscript to make them in a consistent format so that it is easy for the readers to search.
3. Line 350, "value" should be plural here.
4. Line 427, “by” can be changed to “using” in this case.
5. Line 492, “by” can be changed to “using” in this case.
6. Line 496, “structure” can be changed to “microstructure” in this case.
7. Line 499, “as well as” should be changed to “and” in this case.